# Strength of Coarse-Grained Soil Stabilized by Poly (Vinyl Alcohol) Solution and Silica Fume under Wet–Dry Cycles

**DOI:** 10.3390/polym14173555

**Published:** 2022-08-29

**Authors:** Zhewei Zhao, Wenwei Li, Haiping Shi, Zhongyao Li, Jiahuan Li, Cheng Zhao, Peiqing Wang

**Affiliations:** 1Water Conservancy Civil Engineering College, Tibet Agriculture and Animal Husbandry University, Linzhi 860000, China; 2Tibet Construction Water Conservancy and Electric Power Engineering Technology Research Center, Linzhi 860000, China; 3Laboratory of Ministry of Education for Geomechanics and Embankment Engineering, Hohai University, Nanjing 210098, China; 4Key Laboratory of Geotechnical and Underground Engineering of Ministry of Education, Tongji University, Shanghai 200092, China

**Keywords:** coarse-grained soil, PVA, silica fume, wet–dry cycles, unconfined compression strength, stabilization, southeast Tibet

## Abstract

To investigate an environmentally benign stabilizer for coarse-grained soil in southeast Tibet, poly (vinyl alcohol) (PVA) and silica fume were used to improve the geotechnical properties of coarse-grained soil. Unconfined compressive strength (UCS) and wet–dry cycle tests were conducted on prepared samples to evaluate the effect of the additive content and curing age on the strength and durability of coarse-grained soil. The results reveal that the UCS of the samples increased with the additive content of PVA solution and the curing age. The optimal value for the additive content of PVA solution and the curing age is 12% and 7 days, respectively. With the optimal PVA solution content, the PVA solution combined with silica fume stabilizer exhibited better reinforcement compared with pristine PVA. The UCS of the samples stabilized by PVA solution and silica fume increased depending on the curing age, and plateaued after 14 days. Samples with 12% PVA solution and 6% silica fume achieved a satisfactory UCS of 1543.17 kPa after curing for 28 days. As the number of wet–dry cycles increased, the UCS of the samples stabilized by the PVA solution and silica fume exhibited an upward trend during the first three wet–dry cycles, owing to the filling of pores by the gel produced by the silica fume, but began to decline as the number of wet–dry cycles increased. All samples retained a high UCS value after 10 wet-dry cycles compared with the samples that were not subjected to wet–dry cycles.

## 1. Introduction

Under the long-term geological processes of wind and hydraulic forces, many loose coarse-grained soil slopes have formed in southeast Tibet [1,2]. Although coarse-grained soil is characterized by high strength [3], little deformation [4], good hydraulic conductivity [5], and low potential for seismic-induced liquefaction [6], a stable structure is difficult to form between the soil particles, owing to the extremely low cohesion [7], which results in the poor stability of coarse-grained soil slopes under rainfall conditions [8,9]. In recent years, rainfall has been occurring frequently in southeast Tibet as a result of global warming [10,11,12]. Heavy rainfall has triggered many landslides [13,14], which have severely affected the daily lives of local residents and transportation. Therefore, this type of coarse-grained soil slope should be reinforced using a suitable method. However, the altitude of most areas in southeast Tibet exceeds 3000 m and the geological conditions are complex. Reinforcement methods with high disturbance, such as stabilizing piles, retaining walls, compaction, and so on, greatly affect the local environment. Moreover, the reinforcement provided by these methods is easily weakened by the effects of historical earthquakes and the induced resonance coupling effect [15]. Hence, chemical grouting is typically used as a chemical stabilization method for stabilizing coarse-grained soil and reinforcing coarse-grained soil slopes to improve water stability.

With the rapid development of construction technology, chemical stabilization methods [16,17] for improving coarse-grained soil have reached a certain level of maturity. In actual slope engineering, cement [18], lime [19], fly ash [20], and other materials are typically selected as stabilizers by means of grouting [21] to improve the mechanical properties and permeability of coarse-grained soil, and strengthen the landslide mass. However, although the improving effect of traditional cementitious stabilizers is obvious, the addition of such materials causes certain damage to the natural environment [22,23], and does not contribute to the preservation of the ecological environment in Tibet. Moreover, cement production emits a large amount of carbon dioxide and contributes to global warming, which has emerged as a major threat to ecosystems [24]. Therefore, in response to “carbon-neutral” policies, studies are increasingly focusing on new environmentally benign stabilizers. Mombeni et al. [25] investigated the use of lignocellulose micro-fibers (LCMF) as a biological stabilizer for sandy soil by evaluating the performance penetration, shear strength, germination, and water infiltration rate of stabilized samples. Yadav and Tiwari [26] investigated the influence of adding waste rubber tire fibers on the swelling properties, strength, and durability of clay. Zhu et al. [27] combined the addition of hydroxypropyl methylcellulose (HPMC) with MICP technology to improve the soil surface layer. Their test results revealed that the rainwater scour loss and wind erosion loss were reduced.

Poly (vinyl alcohol) (PVA) [28,29] is a type of water-soluble synthetic polymer that can form stronger gels compared with most other synthetic gels. Additionally, PVA is characterized by easy processability, wide commercial availability, nontoxicity, and excellent mechanical properties [30]. Therefore, PVA has been extensively used in various fields, such as composite material preparation and biomedical applications [31]. Owing to its environmentally benign features, the use of PVA as an alternative for cement has attracted significant attention in the field of construction material applications. Yaowarat et al. [32] evaluated the potential of PVA and cement in improving the compressive strength of recycled concrete aggregate. Their results revealed that cement-PVA-stabilized RCA can be used as a supplementary material for complying with road authority requirements. Suksiripattanapong et al. [33] investigated the mechanical and thermal properties of lateritic soil mixed with cement and PVA, and proposed empirical formulas based on experimental results. Silica fume [34,35] is an industrial byproduct with the characteristics of large specific surface area and high activity. Both of these stabilizers have been experimentally shown to improve the geotechnical properties of soil. However, studies on the enhancement of soil strength and water stability by combining PVA and silica flume are scarce, and whether these stabilizers are suitable for coarse-grained soil in high-altitude areas remains an open problem.

This study considered PVA and silica fume as the stabilizer materials, and prepared samples with different proportions, curing time, and wet–dry cycles. The unconfined compressive strength (UCS) was evaluated to investigate the composite stabilizer mechanism so as to improve the strength and water stability of coarse-grained soil. Additionally, the optimal stabilizer combination and content for local coarse-grained soil were established to provide a reference for local coarse-grained soil slope engineering.

## 2. Materials and Methods

### 2.1. Materials

The coarse-grained soil used in the test was taken from the loose deposit beside 318 National Highway in Bomi County, southeastern Tibet Autonomous Region, at a depth of 1.0~1.5 m. After removing impurities from the collected soil samples, they were air-dried, and gray-white coarse-grained soil was obtained for making samples for the unconfined compressive strength test through a 5 mm sieve. Then, geotechnical tests were carried out according to the Standards for Geotechnical Test Methods [36]. The basic physical properties of the soils are listed in Table 1, and the grain size distribution curve is shown in Figure 1. According to the gradation curve, the middle diameter of coarse-grained soil used in the test is long, the proportion of particles over 0.075 mm is greater than 50%, the proportion of particles over 0.5 mm is greater than 5%, the proportion under 0.075 mm is greater than 5% and less than 15%, the coefficient of inhomogeneity Cu is 6.32, and the curvature coefficient Cc is 1.83, Cu > 5 and 1 < Cc < 3. This is fine-grained soil and coarse sand with suitable gradation.

The PVA used in the test has a high degree of polymerization with a 99% degree of alcoholysis. The physical properties are shown in Table 2. Polyvinyl alcohol 20–99 is a white, flaky, featherlike fiber before dissolution that is insoluble in cold water and a stable solution with 5.23% PVA content in boiling water, as shown in Figure 2.

The silica fume used in the test is gray powder with a median diameter of 3.57 µm, average volume diameter of 1.75 µm, and fineness of less than 1 µm, accounting for more than 20%. It has high activity, and its basic physical properties are listed in Table 3.

### 2.2. Experiment Methods

The coarse-grained soil used in the test was air-dried and passed through a 5 mm sieve. Then, impurities such as grass roots, glass, and plastic were removed, and water was added according to its optimum water content. Next, 6, 8, 10 and 12% PVA solution was added to the groups according to the test plan summarized in Table 4. Finally, after mixing uniformly, the mixture was made into cylindrical samples with φ × H = 50 mm × 100 mm by a static compactor. The compaction of the sample was divided into five layers following the requirements for sample preparation in the GB/T50123-2019 Standard for soil test method [36] to achieve uniform compaction in samples.

To better simulate the local environment and actual site conditions of slope engineering in southeastern Tibet, the curing conditions of the samples were close to the natural conditions, which are a temperature of 20 °C and humidity of 40%. Soil samples were stored in a curing chamber for 3, 7, 14 and 28 curing days.

Wet–dry cycle tests were conducted for some of the samples cured for 7 d under natural conditions. The wetting–drying cycle test was divided into two processes: water absorption saturation and air drying. The saturation process of the samples was as follows:(1)Water was gradually added until the sample was completely submerged in water, and the permeable stone was placed at the top of the sample.(2)A certain amount of water was added to the container every 2 h to maintain a constant water level in the container.

The drying process was as follows:(1)The samples were placed in a drying oven after being saturated with water, and the temperature of the drying oven was set to 40 °C.(2)The samples were weighed every 4 h during the drying process to ensure that the moisture content met the test requirements.(3)When the moisture content no longer changed, drying was stopped.

An unconfined compressive strength test was carried out after the sample completed the number of wetting–drying cycles required by the test plan or reached the required curing time to determine the effect of wetting–drying cycles and curing time on the strength of the samples. The test equipment was a fully automatic strain-controlled triaxial apparatus, which can apply a maximum load of 10 kN to the samples. The compression speed was set to 1 mm/min, and the load data applied during the test were recorded to obtain the stress–strain curve and unconfined compressive strength.

According to the UCS value of samples stabilized by different PVA contents, the optimal PVA content and curing time for the coarse-grained soil in southeastern Tibet were determined. Samples with composite stabilizers were made based on the optimal fixed PVA content, and three samples were produced for each ratio. After the preparation of samples, the same curing process and wetting–drying cycles as the previous samples stabilized by PVA alone were carried out. The specific content of silica fume added and the number of wetting–drying cycles are shown in Table 4.

## 3. Results

### 3.1. Improvement Effect of Adding Pristine PVA Solution

Figure 3 shows the samples stabilized by pristine PVA solution after the UCS test. Extremely obvious penetration cracks were observed on the samples, which indicates that pores still existed inside the coarse-grained soil samples. Under the action of external loading, the PVA colloid and colloidal connection between the soil particles broke, and the pores connected to each other and formed penetration cracks. Additionally, the number of cracks observed on the samples subjected to wet–dry cycles is significantly higher. This is attributed to the loss of soil particles, which did not form a monolithic structure with the PVA colloid owing to the multiple wet–dry cycles, thus, the penetration between the pores accelerated.

Figure 4 shows the UCS variation with the curing time for the samples stabilized by pristine PVA solution. The UCS of the samples that cured for three days was not very high because the water content inside the samples was high and some PVA still existed in the form of solution without forming a complete elastic mesh. The strength grew mainly within the curing time of 3–7 days, and the strength increased with the PVA solution content. When the content exceeded 6%, the UCS of the samples that were cured for 7 days reached more than 210% of that of the samples that were cured for 3 days. However, when the curing time exceeded 7 days, the strength growth significantly diminished, which indicates that the PVA colloid had essentially formed. The strength of the non-stabilized soil samples was extremely low, and the UCS was only 10.2 kPa after curing for 7 days, which is much lower compared with the stabilized soil samples.

Figure 5 shows the variation of the UCS with the number of wet–dry cycles for the samples stabilized by pristine PVA solution. As can been seen, the UCS decreased as the number of wet–dry cycles increased. When the PVA solution content was below 12%, each cycle resulted in a strength decrease of 5%–13%, particularly in the first cycle wherein the strength decrease rate was typically more than 10%. When the PVA solution content reached 12%, the strength decrease rate was approximately 4% per cycle, which indicates that the 12% PVA solution addition resulted in the high water stability of the samples. Because coarse-grained soil is cohesionless soil without adhesion between the soil particles, the non-stabilized soil samples essentially disintegrated directly after one wet–dry cycle, and the water stability was extremely poor.

In summary, the strength and water stability of the soil samples stabilized by pristine PVA solution greatly improved compared with non-stabilized soil. By also considering the effect of curing time and the number of wet–dry cycles on the strength of the samples with different PVA solution contents, it is concluded that the samples with 12% PVA solution addition achieved satisfactory strength and water stability, and that the subsequent addition of PVA solution is not cost-effective. For the PVA-stabilized soil samples, the optimal curing age is 7 days.

### 3.2. Improvement Effect of PVA Solution and Silica Fume

Based on the test analysis discussed in the previous section, the addition of PVA solution was set to 12%. The improvement effect of the PVA and silica fume as a composite stabilizer on the coarse-grained soil was further analyzed by adding different silica fume contents.

#### 3.2.1. The Effect of Silica Fume Content on Strength

Figure 6 shows the sample stabilized by the composite stabilizer after the UCS test. Fewer cracks were observed on the sample compared with the samples stabilized by pristine PVA solution. The main change of the sample during the test was the dislodgement of the surrounding soil under external loading, rather than the creation of penetration cracks caused by the fracture of the connections between the soil particles. This phenomenon indicates that the silica fume addition improved the sample’s integrity. The silica fume particles were extremely fine and better filled the pores between the coarse particles, which did not only protect the PVA colloid, but also made the soil structure denser and tightened the occlusion between the soil particles.

Figure 7a shows the change curve of the UCS with the curing time under 12% PVA solution and different silica fume contents. As can be seen, the UCS of the samples increased the most during the natural curing time of 3–7 days, with a growth rate of more than 80%. The growth rate is smaller compared with samples stabilized by pristine PVA solution, because the strength of the sample also improved significantly after 3 days of curing, with a strength increase of 65–90% compared with samples stabilized by pristine PVA solution. This indicates that the silica fume addition accelerated the curing process and filled the pores between the soil particles, which made the sample denser. After curing for more than 7 days, the strength growth rate significantly decreased. With the continuous reduction in the water content, the PVA gradually formed a complete elastic mesh to wrap the soil particles. Moreover, the silica fume gradually reacted and formed a gel that filled the pores between the soil particles and improved the overall strength. The strength essentially reached the peak value after 14 days of curing. For example, after 14 days of curing, the strength of the samples stabilized by 12% PVA solution +6% silica fume was 99% of the strength attained after 28 days of curing.

Figure 7b shows the change curve of the UCS with the silica fume content under the same curing time. As can be seen, as the silica fume content increased, and the UCS of the samples first increased and then decreased. The strength of the samples with 12% PVA solution +6% silica fume was the highest under all curing ages, and reached 1543.17 kPa after 28 days of curing. The reason for this is that the silica fume material had low fineness, and the strength was mainly improved by filling. When the silica fume content reached 6%, the pores between the particles were essentially filled, which decreased the generation speed of PVA in the sample and resulted in failure of forming a more complete elastic mesh. Moreover, owing to the high activity of the silica fume, gel was partially produced. The excessive addition of silica fume resulted in the excessive production of gel, which occupied the space of the PVA mesh that provides strength, and thus the strength of the soil structure decreased to some extent.

#### 3.2.2. Effect of Silica Fume Content on Stress–Strain Characteristics

Figure 8 shows the stress–strain relationship of the composite samples stabilized by different silica fume contents; all curves exhibit strain-softening characteristics. The typical stress–strain relationship is shown in Figure 9. The overall stress–strain curve of the samples can be simplified into four stages:(1)Contact stage: active at the axial strain of 0–1%. At this stage, the pores in the sample gradually compacted under the action of axial pressure, initially forming a skeleton, and the stress increased. As the silica fume addition increased, the curve slope became greater.(2)Compaction stage: active at the axial strain of 1–2.8%. At this stage, the particles contacted each other and began to bear the vertical load. The stress value increased rapidly, and the curve increased linearly.(3)Peak stage: active at the axial strain of 2.8–3.5%. The soil particles began to displace owing to mutual extrusion, the stress growth continued to decrease and reached the peak, and cracks and soil block spalling were observed at the top of the sample.(4)Failure stage: active after the axial strain of 3.5%. As the load borne by the soil skeleton reached the limit, the displacement of soil particles increased further, and the cracks gradually penetrated the entire sample, resulting in rapid strength decline.

The peak secant modulus of the samples was calculated when the specimens reached their peak, and the results are summarized in Table 5.

As can be seen, the peak secant modulus first increased and then decreased as the silica fume content increased, which is similar to the growth of UCS and is approximately hyperbolic with the growth of the curing time. Therefore, the relationship between the three can be fitted by the following empirical formula:(1)$$E=\left(1-\exp\left(a\cdot y\right)\right)\left(b\cdot x+c\cdot x^{2}+d\cdot x^{3}\right)+e$$
where *x* is the silica fume content, *y* is the curing time, and *E* is the peak secant modulus; *a*, *b*, *c*, *d*, and *e* are empirical parameters. For the stabilized coarse-grained soil considered in this study, it can be taken that *a* = −0.3534, *b* = 272.7, *c* = −38.04, *d* = 1.628, and *e* = −222.6. With these parameter settings, the R^2^ for the formula is 0.96, indicating that the predictability is acceptable.

#### 3.2.3. The Effect of Wet–Dry Cycles on Strength

Figure 10 shows the samples with different silica fume contents subjected to 10 wet–dry cycles after the UCS test. As can be clearly seen, when the silica fume content was less than 6%, an obvious Y-shaped crack appeared at the periphery of damaged samples. The reason for this is the presence of unfilled pores inside the sample at low silica fume content, which provided channels for moisture flow and evaporation. The water content at the periphery of the sample changed the most during multiple wet–dry cycles; the soil structure deteriorated, and shrinkage cracks appeared at the periphery of the sample during the drying process. As the silica fume content increased, the number and size of cracks on the surface of damaged samples decreased, but the particle flaking became severe. The integrity of damaged samples improved with the addition of 6% silica fume content, and the spalling of particles and generation of cracks decreased.

Owing to the addition of silica fume, the overall compactness of the composite stabilized samples was excellent. Therefore, during the wet–dry cycle test, the surface layer of the sample did not peel or crack, and the water stability was satisfactory. Figure 11 shows the change trends for the UCS of the composite stabilized samples with the number of wet–dry cycles.

The change trends for the UCS of samples with the number of wet–dry cycles are essentially the same. As the number of wet–dry cycles increased, the UCS of the samples first increased and then decreased. In the process of 1–3 wet–dry cycles, the strength exhibited an upward trend, and then began to decline as the number of wet–dry cycles increased. The overall strength reached the peak after the third wet–dry cycle.

The curve slope in the rising section and curve slope in the falling section were calculated, respectively. It was found that the slope in the rising section was significantly higher than that in the falling section, which means that a small number of wet–dry cycles can significantly improve the strength of composite stabilized samples. With the addition of 4%, 6%, 8%, and 10% silica fume, after three wet–dry cycles, the strength of the samples increased by 51.72%, 39.75%, 44.78%, and 45.03%, respectively, compared with the samples subjected to natural curing for 7 days. Although the strength of the samples decreased after several wet–dry cycles, the decrease was obviously small. After 10 wet–dry cycles, the strength of the samples with 4%, 6%, 8%, and 10% silica fume addition exceeded 1 MPa, and the strength decreased by 3.70%, 1.32%, 14.05%, and 12.24% compared with the samples subjected to natural curing for 28 days. The overall strength attenuation was within 15% of the natural curing peak strength.

#### 3.2.4. Microscopic Characteristics of Samples Stabilized by PVA Solution and Silica Fume

To visualize the effect of silica fume and PVA solution on the strength and water stability of the coarse-grained soil, the microstructures were observed by scanning electron microscopy. Figure 12 shows the microstructure of the samples stabilized by PVA solution and silica fume after 3, 7, and 28 days of curing. As shown in Figure 12a, a mesh-like structure was not observed and the gels were not adequately integrated, which led to the UCS of all samples being low at the curing age of 3 days. As the curing age increased, obvious gels were observed, as shown in Figure 12b,c, and these gels formed a more complete elastic three-dimensional mesh, while the silica fume particles filled the pores. This resulted in a dense structure consisting of the trio “silica-fume-PVA mesh-soil particles”, which greatly improved the strength of the samples.

Figure 13 shows the microstructure of the samples stabilized by PVA solution and silica fume after 10 wet–dry cycles with the silica fume contents of 4%, 6%, and 8%. As shown in Figure 13a, after 10 wet–dry cycles, the soil structure exhibited different degrees of deterioration depending on the silica fume content. When the silica fume content was 4%, the generated silica fume gel failed to adequately fill the pores between the soil particles after several wet–dry cycles, and various pores existed between the soil particles. As the silica fume content increased, the pores in the soil structure decreased as shown in Figure 13b, and the dense structure formed by the silica-fume-PVA mesh-soil particles still had excellent integrity after several wet–dry cycles. However, when the silica fume content exceeded 6%, the lower-strength silica fume gel squeezed the space of the PVA elastic mesh and even broke the connection between the PVA gel and the soil particles, as shown in Figure 13c. This led to the generation of micro cracks, which suggests that excessive silica fume can lead to strength reduction.

#### 3.2.5. The Mechanism of Strength and Water Stability Enhancement of the Coarse-Grained Soils Stabilized with PVA and Silica Fume

Figure 14 shows a simple schematic diagram of the strength mechanism and water stability enhancement of the coarse-grained soils with the addition of PVA solution and silica fume. In the wet–dry cycle test, the overall strength of the composite-stabilized samples first increased and then decreased. The main reason for this is that when the composite-stabilized samples cured for 7 days, most of the silica fume did not react and was still filled between the pores in the form of fine particles, which limited the improvement of soil strength. During the wet–dry cycles, the full saturation of the samples and the suitable temperature conditions during air drying assisted the stimulation of the silica fume’s own activity and the production of gels that could effectively fill the pores of particles and better combine with the elastic mesh generated by the PVA, which improved the compressive strength of the samples. With the further increase in the number of wet–dry cycles and the loss of particles, the gel deteriorated to a certain extent, but could still maintain an excellent and stable structure with the soil particles.

However, when the silica fume content exceeded 6%, the gel formed by the excessive silica fume squeezed the space of the PVA colloid and caused damage to the elastic mesh, and the connecting body between the particles changed from high-strength PVA to silica fume colloid with poor cohesion. Moreover, the silica fume gel overflowing the pores damaged the original dense structure, which also increased the particle loss and decreased the strength in subsequent wet–dry cycles.

## 4. Conclusions

This study conducted unconfined compressive testing to investigate the effect of the stabilizer content, curing time, and wet–dry cycles on the mechanical properties of coarse-grained soil stabilized by PVA solution and silica fume. For pristine-PVA-solution-stabilized samples, high strength and water stability were achieved with the addition of 12% PVA solution, and the optimal curing age was 7 days. For samples stabilized by PVA solution and silica fume, the strength increased significantly within 3–7 days and then tended to be stable after 14 days of curing. The improvement effect of 12% PVA solution +6% silica fume was optimal, and the peak strength reached 1543.17 kPa under natural curing conditions. During the wet–dry cycles, the strength increased in 1–3 wet–dry cycles owing to the full saturation of the samples and beneficial temperature conditions during air drying. This process assisted the stimulation of the silica fume’s own activity and gel production. This led to the effective filling of the particle pores and better combination with the elastic mesh generated by the PVA, which improved the strength of the soil structure. After 10 wet–dry cycles, the strength decreased within 15% of the peak strength under natural curing conditions. The low decrease in strength after multiple wet–dry cycles was caused by the fact that with the further increase in the number of wet–dry cycles accompanied by particle loss, the gel exhibited some deterioration but still maintained a good and stable structure with the soil particles. However, it should be noted that it is unclear whether the uniformity of such composite grouting materials can be guaranteed during the actual construction process. Therefore, studies related to field construction are necessary.

## Figures and Tables

**Figure 1 polymers-14-03555-f001:**
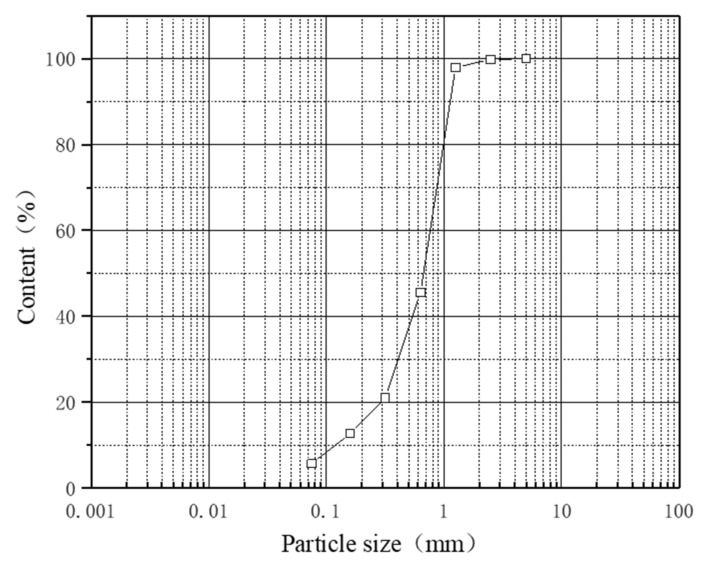
Grain size distribution curve of the coarse-grained soil.

**Figure 2 polymers-14-03555-f002:**
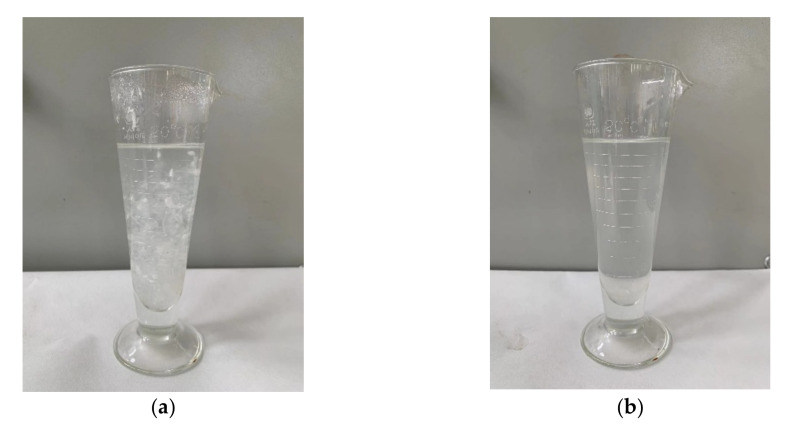
Preparation of stable PVA Solution for (**a**) PVA added and (**b**) PVA dissolution.

**Figure 3 polymers-14-03555-f003:**
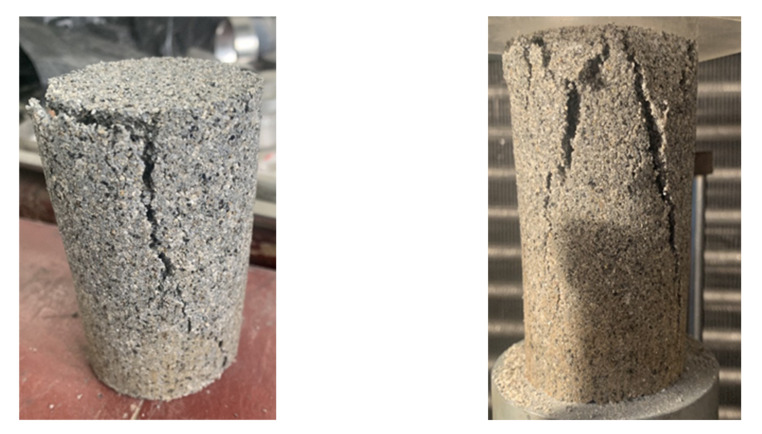
Samples after UCS test for (**a**) without wet–dry cycles and (**b**) with wet–dry cycles.

**Figure 4 polymers-14-03555-f004:**
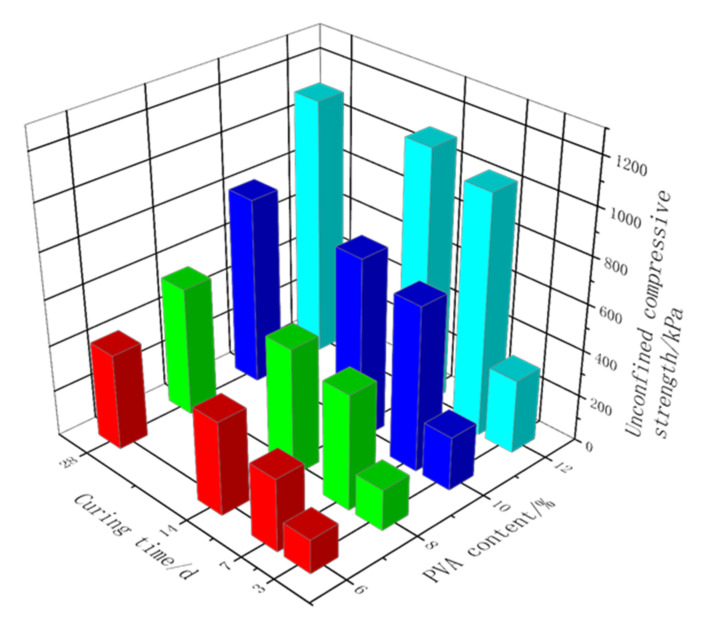
Statistical histogram of UCS of coarse-grained soil samples stabilized by PVA solution under different curing times.

**Figure 5 polymers-14-03555-f005:**
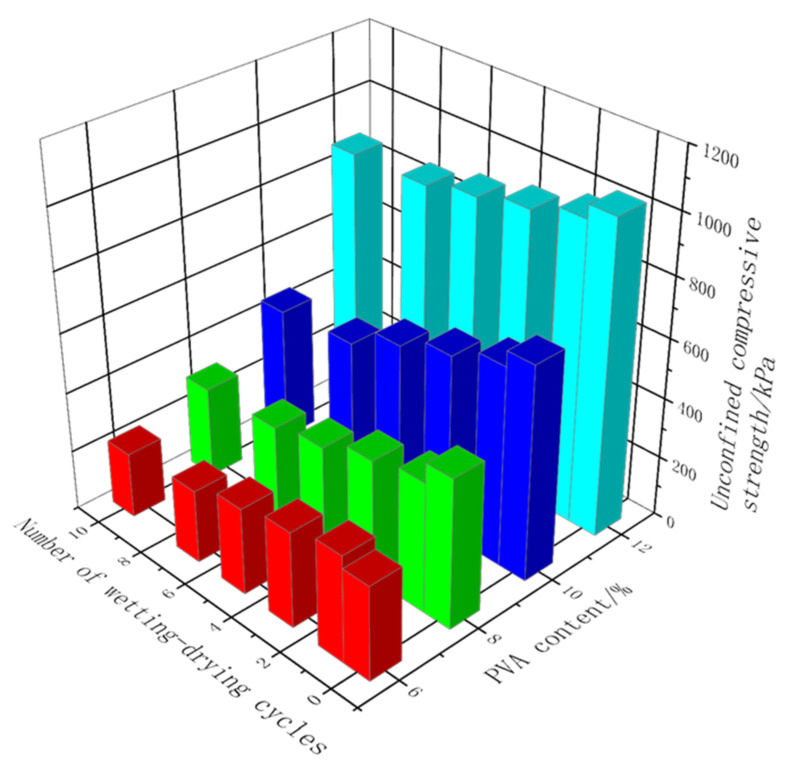
Statistical histogram of UCS of coarse-grained soil samples stabilized by pristine PVA solution after different numbers of wet–dry cycles.

**Figure 6 polymers-14-03555-f006:**
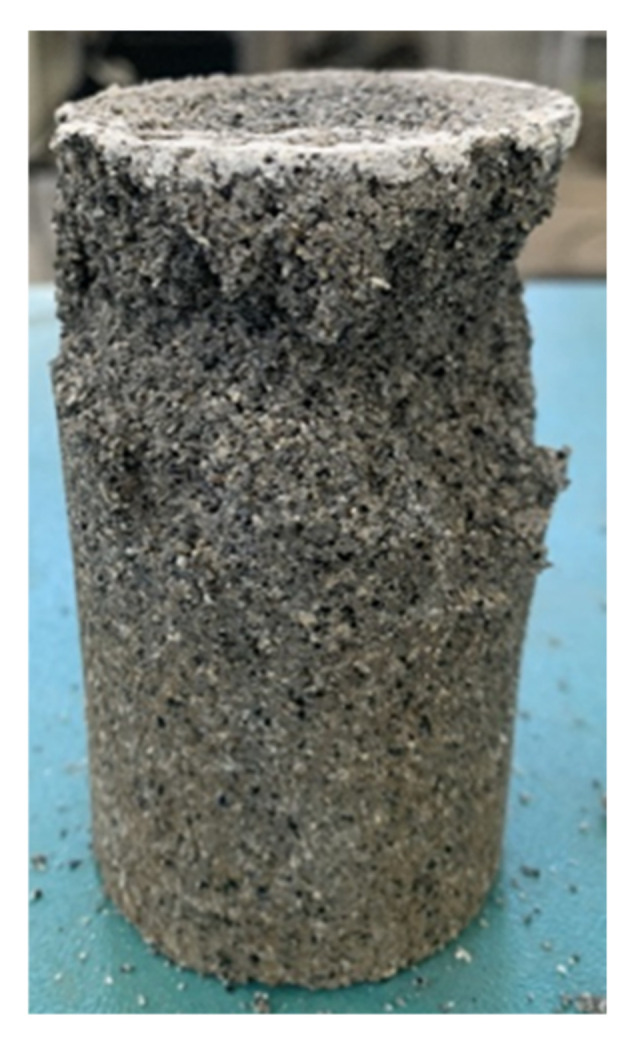
Sample stabilized by composite stabilizer after UCS test.

**Figure 7 polymers-14-03555-f007:**
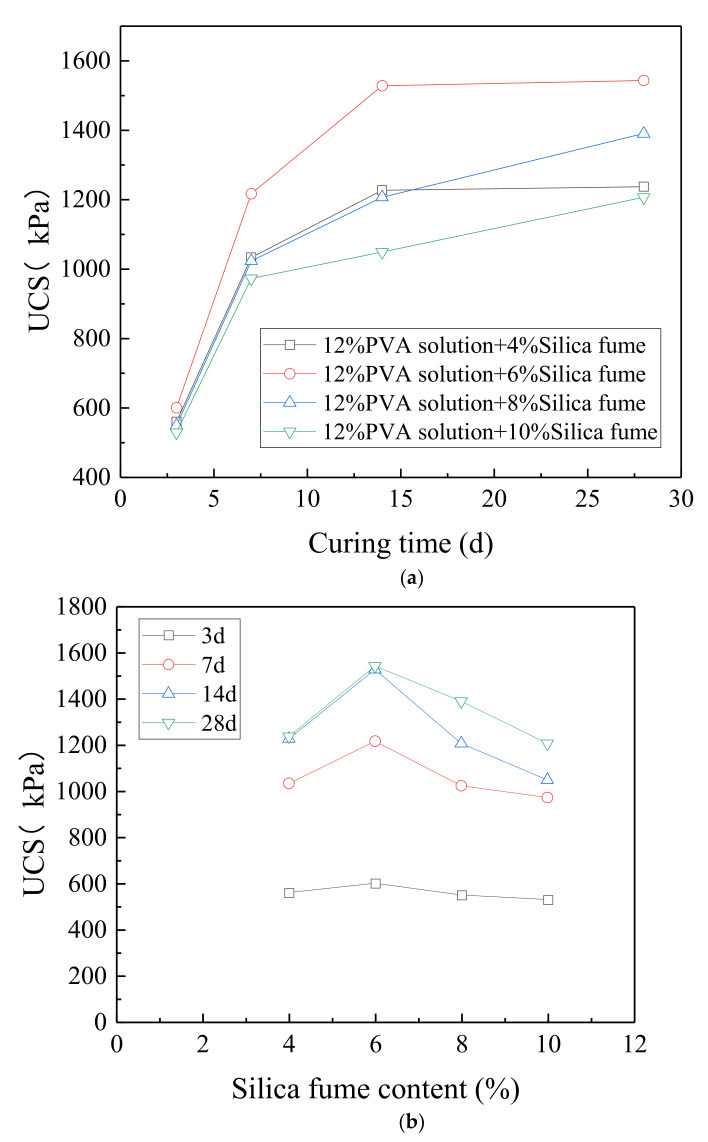
Relationship between (**a**) curing time, (**b**) silica fume content, and UCS of stabilized samples.

**Figure 8 polymers-14-03555-f008:**
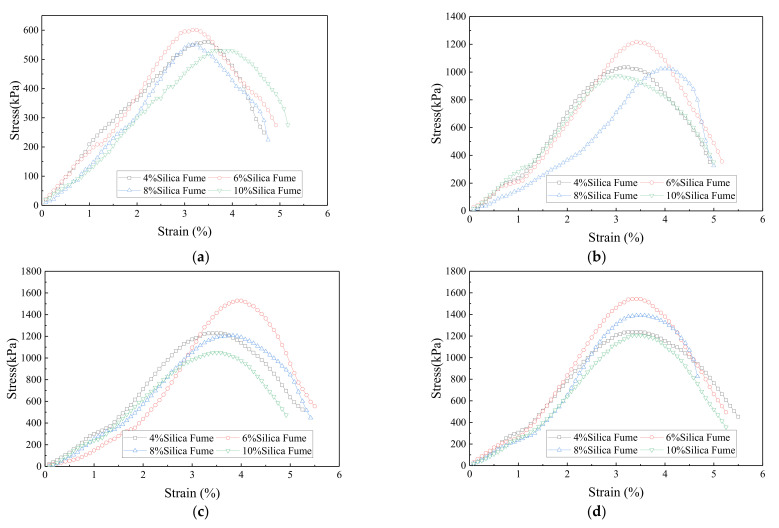
Stress–strain relationship of stabilized soil samples with different silica fume contents for different curing time: (**a**) 3d; (**b**) 7d; (**c**) 14d; (**d**) 28d.

**Figure 9 polymers-14-03555-f009:**
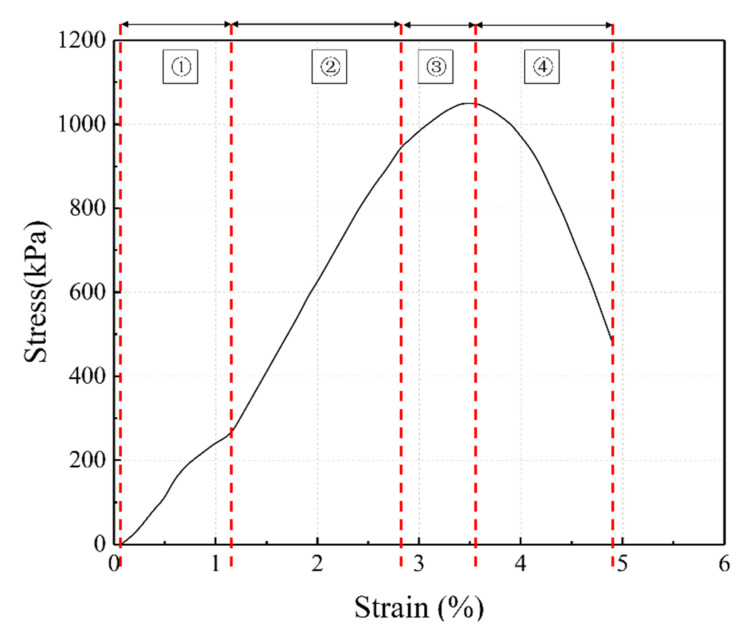
Schematic diagram of typical stress–strain curve.

**Figure 10 polymers-14-03555-f010:**
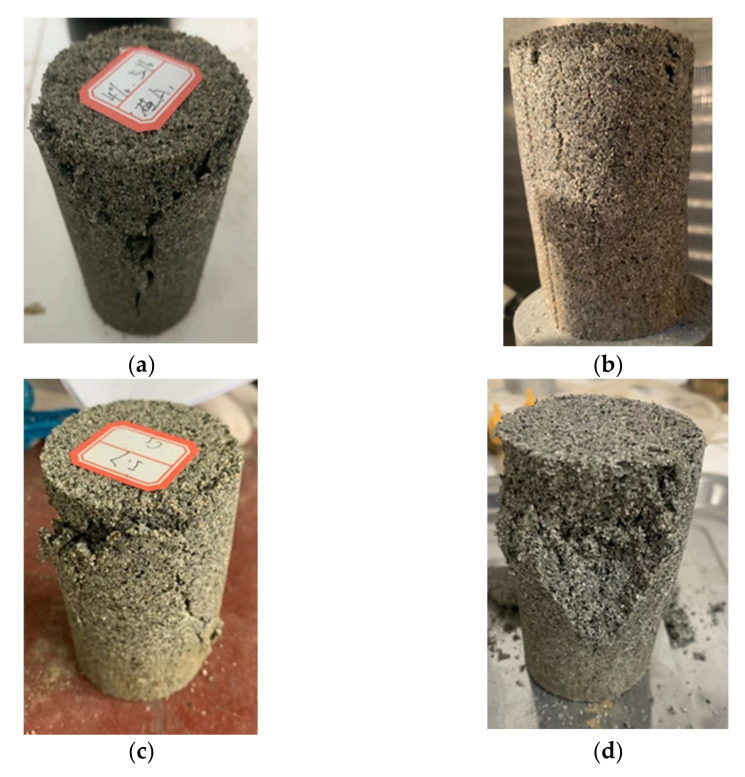
Composite samples with different silica fume contents subjected to 10 wet–dry cycles after UCS test: (**a**) 4%; (**b**) 6%; (**c**) 8%; (**d**) 10%.

**Figure 11 polymers-14-03555-f011:**
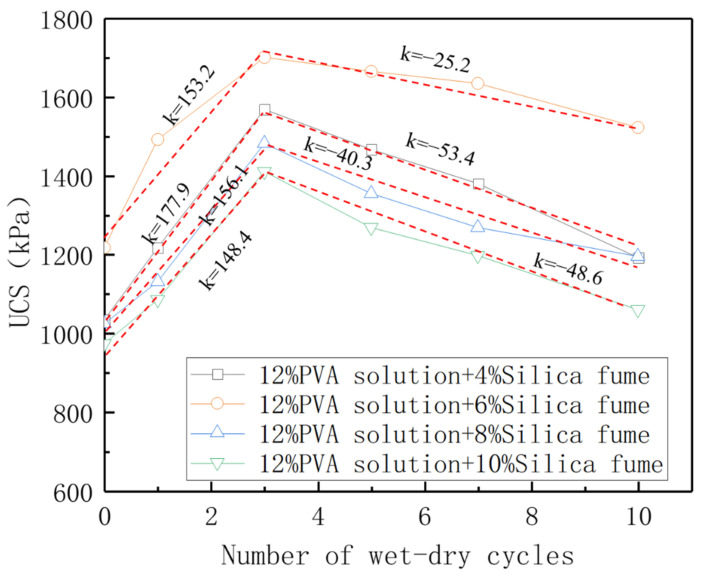
Change curve of UCS with number of wet–dry cycles.

**Figure 12 polymers-14-03555-f012:**
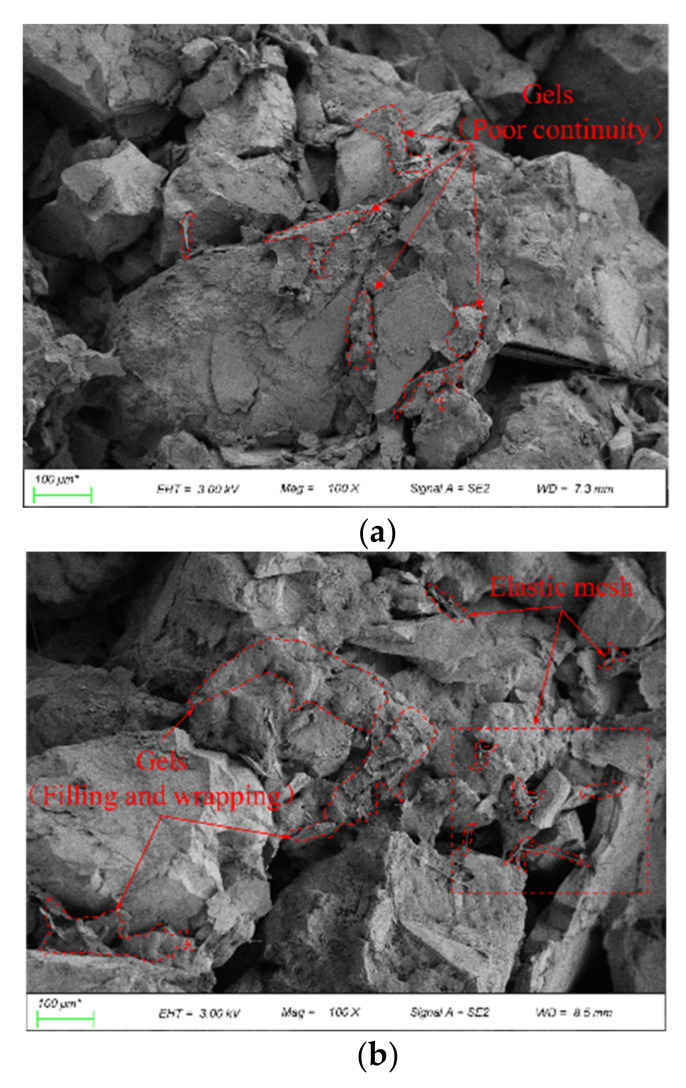
The microstructure of the samples stabilized by PVA solution and silica fume after (**a**) 3 days, (**b**) 7 days, and (**c**) 28 days of curing.

**Figure 13 polymers-14-03555-f013:**
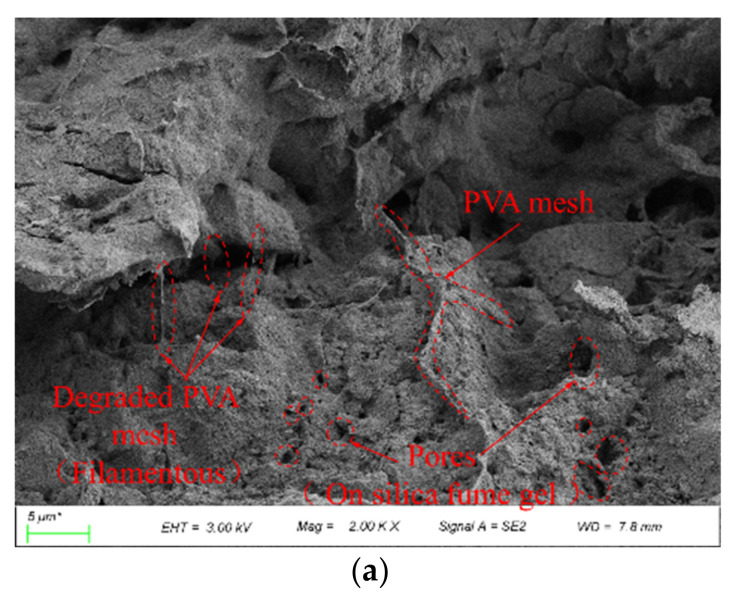
The microstructure of the samples stabilized by PVA and silica fume after 10 wet–dry cycles (**a**) 4% (**b**) 6% (**c**) 8%.

**Figure 14 polymers-14-03555-f014:**
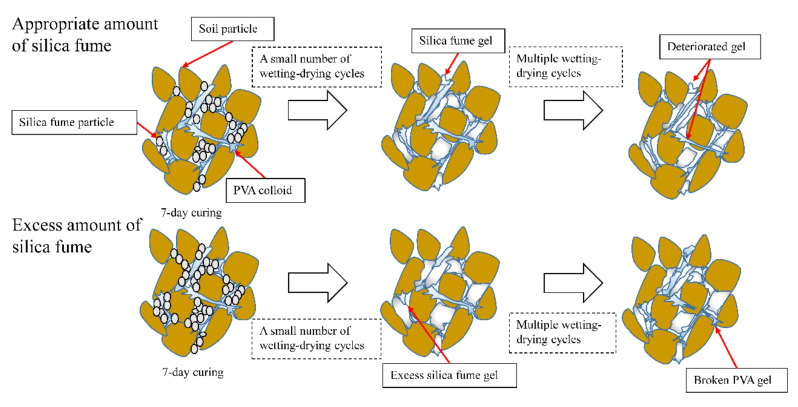
Schematic diagram of influence mechanism of wet–dry cycles on strength of composite-stabilized sample.

**Table 1 polymers-14-03555-t001:** Basic physical properties of coarse-grained soils for the test.

Optimum Moisture Content (%)	Maximum Dry Density (g/cm^3^)	d_10_ (mm)	d_60_ (mm)	Cu	Cc
12	1.79	0.13	0.80	6.32	1.83

**Table 2 polymers-14-03555-t002:** Basic properties of PVA for testing.

Viscosity (mPas)	Volatile Components (%)	Ash Content (%)	PH
34.0–42.2	5	0.5	5–7

**Table 3 polymers-14-03555-t003:** Basic properties of silica fume used in the test.

Fire Resistance (°C)	Volume Weight (kg/m^3^)	Average Particle Size (μm)	Specific Surface Area (m^2^/kg)
>1700	220–250	3.4–3.6	1262.85

**Table 4 polymers-14-03555-t004:** The stabilizer content, curing time and number of wet–dry cycles adopted for different samples.

Group	Number	PVA Solution Content (%)	Silica Fume Content (%)	Curing Time (d)	Number of Wet–Dry Cycles
**Non-stabilized Soil**	1	0	0	3,7,14,28	0
	2	0	0	7	1
**Pristine PVA Solution**	3	6	0	3,7,14,28	0
	4	6	0	7	1,3,5,7,10
	5	8	0	3,7,14,28	0
	6	8	0	7	1,3,5,7,10
	7	10	0	3,7,14,28	0
	8	10	0	7	1,3,5,7,10
	9	12	0	3,7,14,28	0
	10	12	0	7	1,3,5,7,10
**PVA Solution and Silica Fume**	11	12	4	3,7,14,28	0
	12	12	4	7	1,3,5,7,10
	13	12	6	3,7,14,28	0
	14	12	6	7	1,3,5,7,10
	15	12	8	3,7,14,28	0
	16	12	8	7	1,3,5,7,10
	17	12	10	3,7,14,28	0
	18	12	10	7	1,3,5,7,10

**Table 5 polymers-14-03555-t005:** The peak secant modulus for the samples.

	Curing Time(d)			
Silica Fume Content (%)	3	7	14	28
4	160.1	318.1	342.5	353.6
6	184.9	356.3	382.0	440.9
8	169.2	261.4	314.9	397.3
10	132.4	315.5	292.8	344.9

## Data Availability

This will be made available upon request through the corresponding author.

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
