# Peer review of "Strength of Coarse-Grained Soil Stabilized by Poly (Vinyl Alcohol) Solution and Silica Fume under Wet–Dry Cycles"

_polymers, 2022, doi:10.3390/polym14173555_

Round 1
Reviewer 1 Report
In the title, each word's first character should be capitalised.
In correspondence please indicate who is the corresponding author in the common bracket after the email.
Line 23, ‘alone’ please change it to pristine PVA or pure PVA.
Line 37 there should be space words and citations. Forex: space should be conductivity[5] check another place also.
Write polyvinyl alcohol as ‘poly (vinyl alcohol)’ throughout the manuscript. The introduction needs to written more deeply considering the PVA materials' chacterstices. The following articles are cited accordingly in the introduction. https://doi.org/10.1080/10298436.2020.1746311, https://doi.org/10.1007/s10924-022-02454-w; https://doi.org/10.1016/j.cscm.2022.e00962, Carbohydrate Polymers (IF 9.38), 257 (2021) 117633.
Once you define short name of any terminology, use that short name throughout the manuscript. For ex: in line polyvinyl alcohol should PVA. Check throughout the manuscript.
Figure 3 is cited first in the text and figure 2 is later so change the figure number.
Provide the mechanism, why coarse grained soil is stabilized with PVA and silica fume.
Line 108, what is um it should be micro symbol. Check throughout the manuscript.
There are many figures in the manuscript please make it subfigure and reduce it to 7-8 figures.
Table 4 caption needs to be rewritten. Write in commas in the proper way.
Fig 13 and 17 Clarity is so low improve it.
In conclusion section need to rewritten. Avoid pointwise sentence make it just one paragraph.
Check the reference style of the polymers journal. Polymers journal need journal abbreviation. Please double-check it.
English need to be corrected throughout the manuscript. So many grammatical errors.
Author Response
Dear Editor and Reviewers:
On behalf of my co-authors, we are very grateful to you for giving us an opportunity to revise our manuscript. We appreciate you very much for your positive and constructive comments and suggestions on our manuscript entitled “Strength of coarse-grained soil stabilized by polyvinyl alcohol and silica fume under wet–dry cycles” (ID: polymers-1811165).
We have studied reviewers’ comments carefully and tried our best to revise our manuscript according to the comments. The changes in the revised manuscript can be seen by Microsoft Word’s Track Changes feature. The following are the responses and revisions I have made in response to the reviewers' questions and suggestions on an item-by-item basis. Thanks again to the hard work of the editor and reviewers!
Response to the comments of Reviewer #1
Comment No. 1: In the title, each word's first character should be capitalized.
Response: In the revised manuscript, the title is changed to “Strength of Coarse-Grained Soil Stabilized by Polyvinyl Alcohol Solution and Silica Fume under Wet–Dry Cycles”.
Comment No. 2: In correspondence please indicate who is the corresponding author in the common bracket after the email.
Response: Thanks for the suggestion, in the revised manuscript, the name for the corresponding author is indicated in the common bracket after the email.
Comment No. 3: Line 23, ‘alone’ please change it to pristine PVA or pure PVA.
Response: Thanks for the suggestion, in the revised manuscript, the ‘PVA alone’ has been changed to ‘pristine PVA’.
Comment No. 4: Line 37 there should be space words and citations. Forex: space should be conductivity[5] check another place also.
Response: In the revised manuscript, the space words have been supplemented.
Comment No. 5: Write polyvinyl alcohol as ‘poly (vinyl alcohol)’ throughout the manuscript. The introduction needs to written more deeply considering the PVA materials' chacterstices. The following articles are cited accordingly in the introduction. https://doi.org/10.1080/10298436.2020.1746311, https://doi.org/10.1007/s10924-022-02454-w; https://doi.org/10.1016/j.cscm.2022.e00962, Carbohydrate Polymers (IF 9.38), 257 (2021) 117633.
Response: (1) In the revised manuscript, polyvinyl alcohol has been changed to ‘poly (vinyl alcohol)’. (2) These articles are meaningful for understanding the PVA materials' characteristics and the content has been supplemented in the introduction.
Comment No. 5: Once you define short name of any terminology, use that short name throughout the manuscript. For ex: in line polyvinyl alcohol should PVA. Check throughout the manuscript.
Response: Thanks for the suggestion, in the revised manuscript, the short name has been used.
Comment No. 6: Figure 3 is cited first in the text and figure 2 is later so change the figure number.
Response: In the revised manuscript, the number of figures is reduced and the remaining figures are renumbered.
Comment No. 7: Provide the mechanism, why coarse grained soil is stabilized with PVA and silica fume.
Response: Considering the Reviewer’s suggestion, the sections of the revised manuscript have been adjusted. Section 3.2.4 is supplemented to explain the enhancement mechanism of strength and water stability of the coarse-grained soils stabilized with PVA solution and silica fume.
Comment No. 8: Line 108, what is um it should be micro symbol. Check throughout the manuscript.
Response: We are sorry that due to our negligence, the unit is misused. In the revised manuscript, all units have been checked
Comment No. 9: There are many figures in the manuscript please make it subfigure and reduce it to 7-8 figures.
Response: As reviewer suggested, we have reduced the number of figures. However, to ensure that the results can be displayed completely, the number of figures is still more than 8.
Comment No. 10: Table 4 caption needs to be rewritten. Write in commas in the proper way.
Response: As reviewer suggested, the Table 4 caption has been changed to ‘The stabilizer content, curing time and number of wet-dry cycles adopted for different samples’
Comment No. 11: Fig 13 and 17 Clarity is so low improve it.
Response: As reviewer suggested, the clarity of Fig 13 and 17 has been improved.
Comment No. 12: In conclusion section need to rewritten. Avoid pointwise sentence make it just one paragraph.
Response: As reviewer suggested, the conclusion section has been rewritten.
Comment No. 13: Check the reference style of the polymers journal. Polymers journal need journal abbreviation. Please double-check it.
Response: The style of references have been corrected in the revised manuscript.
Comment No. 14: English need to be corrected throughout the manuscript. So many grammatical errors.
Response: As reviewer suggested, the English has been corrected throughout the manuscript.

Reviewer 2 Report
This research tries to understand the long term strength of stabilized coarse grained soils. However, the purpose of performing durability cycles on coarse grained soils with silica fumes and PVC stabilizer is not convincing. Compared to fine grained soils, coarse grained are more stable and do not exhibit swell/shrink characteristics with wet/dry conditions. Soil slope failures as identified in sentences 35-40, would have occurred due to poor slope conditions, gravity or heavy rainfall conditions. We understand that coarse grained soils are not a slope fill material. But, if we find a case where there exists a slope with coarse grained material, what should we do is the question. What are the authors recommending to stabilize such slopes?
Some other comments:
Line 119 - What AASHTO or other standard procedure is used for achieving uniform compaction in specimens?
Line 383 - 12% PVC with 6% Silica Fume is a very high amount of stabilizer dosage. Did the authors perform a volumetric cost analysis for stabilizing this material. Would rock anchors or other ground improvement methods be cheaper and more effective?
Overall, good experimental findings. However, the lack of project need and purpose diminishes the overall quality. I recommend authors to focus on project need and purpose and identify the methods or practices that would help retrofit existing slopes on coarse grained material. Is this type of stabilization feasible for soil slopes in Southeast Tibet?
Author Response
Dear Editor and Reviewers:
On behalf of my co-authors, we are very grateful to you for giving us an opportunity to revise our manuscript. We appreciate you very much for your positive and constructive comments and suggestions on our manuscript entitled “Strength of coarse-grained soil stabilized by polyvinyl alcohol and silica fume under wet–dry cycles” (ID: polymers-1811165).
We have studied reviewers’ comments carefully and tried our best to revise our manuscript according to the comments. The changes in the revised manuscript can be seen by Microsoft Word’s Track Changes feature. The following are the responses and revisions I have made in response to the reviewers' questions and suggestions on an item-by-item basis. Thanks again to the hard work of the editor and reviewers!
Response to the comments of Reviewer #2
Comment No. 1: This research tries to understand the long term strength of stabilized coarse grained soils. However, the purpose of performing durability cycles on coarse grained soils with silica fumes and PVC stabilizer is not convincing. Compared to fine grained soils, coarse grained are more stable and do not exhibit swell/shrink characteristics with wet/dry conditions. Soil slope failures as identified in sentences 35-40, would have occurred due to poor slope conditions, gravity or heavy rainfall conditions. We understand that coarse grained soils are not a slope fill material. But, if we find a case where there exists a slope with coarse grained material, what should we do is the question. What are the authors recommending to stabilize such slopes?
Response: Although coarse grained soil are more stable and do not exhibit swell/shrink characteristics with wet/dry conditions, the durability and water stability of coarse grained soils is still needed to improve for slope engineering. Based on our pre-experiments, the coarse grained soil samples is easy to disintegrate and soil particles are easy to lose during saturation. When the water evaporates, the cohesionless characteristic of coarse-grained soils will cause the soil to loosen. With the continuous loss of particles and loosening of structure, the strength of coarse grained soil will be reduced significantly. In actual slope engineering, the low durability and water stability will easily lead to landslides which were reported many times in Southeast Tibet. In order to avoid landslides, grouting is usually used to reinforce the coarse grained soils of slope in Tibet, so this paper focuses on exploring a binding material which can be used in grouting technique to provide an alternative for cement. The experiment results can be a reference for slope grouting reinforcement engineering in southeast Tibet. In the revised manuscript, we have added content about project need and purpose in the introduction to focus on the applicability for chemical grouting technique that would help retrofit existing slopes on coarse grained material.
Comment No. 2: Line 119 - What AASHTO or other standard procedure is used for achieving uniform compaction in specimens?
Response: For sample preparation, “GB/T50123-2019 Standard for soil test method” is used to achieve uniform compaction in specimens. According to the requirements for sample preparation in unconfined compressive test, the compaction of the sample is divided into five layers. The soil mass of each layer is equal based on the density used in the test. After each layer is compacted to the required height, the surface is roughened and a new layer of soil is added. Repeat compaction and adding soil until the last layer of soil is compacted, and a sample with uniform compaction is completely made. To help readers better know about the process, the process has been briefly introduced and the “GB/T50123-2019 Standard for soil test method” is cited.
Comment No. 2: Line 383 - 12% PVC with 6% Silica Fume is a very high amount of stabilizer dosage. Did the authors perform a volumetric cost analysis for stabilizing this material. Would rock anchors or other ground improvement methods be cheaper and more effective?
Response: (1) We are sorry that the stabilizers were not illustrated clearly, leading the amount of stabilizer dosage was misunderstood. The percentage “12%”is for the PVA solution instead of the pristine PVA. The stable solution contains 5.23% PVA in boiling water. Hence, the real percentage for the PVA added is 0.64%, which is similar with other researches. To avoid the misunderstanding, the “PVA” is changed to “PVA solution”. (2) The percentage of PVA and Silica Fume added is determined by pre-experiments about strength and the volumetric cost analysis. Based on the results of pre-experiments, the strength of samples with low percentage stabilizers (PVA solution <3% and silica fume <2%) is extremely low. And most of samples fail to maintain integrity after 1-2 wet-dry cycles. Hence, the amount of stabilizer dosage used in our research is 6-12% and 4-10% for PVA solution and silica fume, respectively. (3) Our research focus on exploring a binding material which can be used in grouting technique to provide an alternative for cement. The cost performance and effectiveness of ground improvement methods are not the focus of our research. In fact, the geological conditions of southeast Tibet is complex, and the cost performance and effectiveness should be evaluated based on the specific engineering conditions.
Comment No. 3: Overall, good experimental findings. However, the lack of project need and purpose diminishes the overall quality. I recommend authors to focus on project need and purpose and identify the methods or practices that would help retrofit existing slopes on coarse grained material. Is this type of stabilization feasible for soil slopes in Southeast Tibet?
Response: As the reviewer suggested, we have added content about project need and purpose in the introduction to focus on the applicability for chemical stabilization methods that would help retrofit existing slopes on coarse grained material.

Reviewer 3 Report
Results on the stabilization of coarse-grained soil by PVA/SiO2 were presented in this manuscript. The finding may have significant practical applications; however, I think that the manuscript is better suited for another journal (e.g. Materials MDPI). In fact the only polymer-related aspect in this manuscript is the use of poly(vinyl alcohol).
Moreover, I think that the figures should be revised. Figure 2 does not show solubility of PVA in water as claimed in line 102. Figures 1, 3, 4 5, 6, 7must be removed – they are completely useless as an illustration.
Author Response
Dear Editor and Reviewers:
On behalf of my co-authors, we are very grateful to you for giving us an opportunity to revise our manuscript. We appreciate you very much for your positive and constructive comments and suggestions on our manuscript entitled “Strength of coarse-grained soil stabilized by polyvinyl alcohol and silica fume under wet–dry cycles” (ID: polymers-1811165).
We have studied reviewers’ comments carefully and tried our best to revise our manuscript according to the comments. The changes in the revised manuscript can be seen by Microsoft Word’s Track Changes feature. The following are the responses and revisions I have made in response to the reviewers' questions and suggestions on an item-by-item basis. Thanks again to the hard work of the editor and reviewers!
Response to the comments of Reviewer #3
Comment No. 1: Results on the stabilization of coarse-grained soil by PVA/SiO2 were presented in this manuscript. The finding may have significant practical applications; however, I think that the manuscript is better suited for another journal (e.g. Materials MDPI). In fact the only polymer-related aspect in this manuscript is the use of poly (vinyl alcohol).
Response: Thanks for the suggestion, this manuscript focus on the strength and durability of pristine Poly (Vinyl Alcohol) stabilized coarse-grained soil and the enhancement of silica fume combined with PVA. PVA is one kind of water-soluble synthetic polymer which is derived from poly (vinyl acetate) through alkaline hydrolysis. Meanwhile, some similar research papers were published in the journal. Hence, we still hope to submit this manuscript to the journal Polymers.
Comment No. 2: Moreover, I think that the figures should be revised. Figure 2 does not show solubility of PVA in water as claimed in line 102. Figures 1, 3, 4 5, 6, 7must be removed – they are completely useless as an illustration.
Response: In the revised manuscript, the clarity and content of figures have been improved. The number of figures is also reduced.

Round 2
Reviewer 1 Report
The authors improved the manuscript. Now It can publish as such.
Author Response
I would like to thank the reviewers for their comments. These comments have improved the quality of the manuscript.
Reviewer 2 Report
Manuscript reads a lot better from the previous version. However, some minor changes are still needed
1. Authors have mentioned chemical grouting technique in the literature (provide references to lines 54-56). However, no bridge between this research and grouting technique has been presented. At the end of conclusion section, I recommend the authors to add some context regarding application of PVC stabilizer as a grouting method and the challenges during construction (transporting, application, curing etc).
2. I recommend the authors to combine all figures regarding sample preparation to 1. Figures like 12 and 13 need to be blown up as it is hard to see the SEM pictures which are compressed. Figure 14 is not clear.
3. Line 503 - Avoid using terms like "good" as we cannot define these relatively
4. Lines 194-200 are redundant, please compress these to one line. Line 214 - How did you determine that there is no change in moisture. Until what decimal did you take the weight readings?
5. Line 393- What is the predictability of Equation 1 presented. Can the authors present the R-square value when compared with measured data?
6. After 10 wetting drying cycles, most of the samples lost their strength and returned to their 15% of original peak. Why? Also, can the authors relate 10 w/d cycles to how many years in actual conditions. Does this mean this stabilizer need to be periodically applied to slopes and is not a permanent solution. If so, please highlight in conclusions
Author Response
Comment No. 1: Authors have mentioned chemical grouting technique in the literature (provide references to lines 54-56). However, no bridge between this research and grouting technique has been presented. At the end of conclusion section, I recommend the authors to add some context regarding application of PVC stabilizer as a grouting method and the challenges during construction (transporting, application, curing etc).
Response: Considering the reviewer’s suggestions, the challenge that may be faced in the construction are mentioned.
Comment No. 2: I recommend the authors to combine all figures regarding sample preparation to 1. Figures like 12 and 13 need to be blown up as it is hard to see the SEM pictures which are compressed. Figure 14 is not clear.
Response: Considering the reviewer’s suggestions, the figures are modified.
Comment No. 3: Line 503 - Avoid using terms like "good" as we cannot define these relatively.
Response: Considering the reviewer’s suggestions, the terms are replaced.
Comment No. 4: Lines 194-200 are redundant, please compress these to one line. Line 214 - How did you determine that there is no change in moisture. Until what decimal did you take the weight readings?
Response: The redundant content has been compressed. Determine that the moisture has not changed when the weight of the sample is no longer changing. The weight is read to two decimal places.
Comment No. 5: Line 393- What is the predictability of Equation 1 presented. Can the authors present the R-square value when compared with measured data?
Response: The R2 for the formula is 0.96, indicating that the predictability is acceptable.
Comment No. 6: After 10 wetting drying cycles, most of the samples lost their strength and returned to their 15% of original peak. Why? Also, can the authors relate 10 w/d cycles to how many years in actual conditions. Does this mean this stabilizer need to be periodically applied to slopes and is not a permanent solution. If so, please highlight in conclusions.
Response: The reasons for the decrease in strength are introduced in section 3.2.4. With the further increase of the number of wet–dry cycles and the loss of particles, the gel deteriorated to a certain extent, resulting in a small decrease. The decrease in strength is acceptable, because the wet and dry cycle will certainly cause a decrease in strength. One wet-dry cycle can be related to a simulation for 1 year in actual conditions. In this paper, we use rapid wet-dry cycle mainly to study the improvement effect and feasibility of this stabilizer. Based on the results, the long-term strength of the improved soil is acceptable. In fact, the reinforcement of slope can hardly be called a permanent measure. Because the local environment is affected by many factors, rainfall, earthquake, landslide, etc. can weaken the reinforcement effect. The research mainly provides a new idea for the selection of grouting materials for local slope reinforcement projects, and whether PVA needs to be applied regularly will be studied in later work.